# Detecting Change in Seasonal Pattern via Autoencoder and Temporal Regularization

## Abstract

Change-point detection problem consists of discovering abrupt property changes in the generation process of time-series. Most state-of-the-art models are optimizing the power of a kernel two-sample test, with only a few assumptions on the distribution of the data. Unfortunately, because they presume the samples are distributed $i.i.d.$, they are not able to use information about the seasonality of a time-series. In this paper, we present a novel approach - **ATR-CSPD** allowing the detection of changes in the seasonal pattern of a time-series. Our method uses an autoencoder together with a temporal regularization, to learn the pattern of each seasonal cycle. Using low dimensional representation of the seasonal patterns, it is possible to accurately and efficiently estimate the existence of a change point using a clustering algorithm. Through experiments on artificial and real-world data sets, we demonstrate the usefulness of the proposed method for several applications.

## 1 Introduction

Time series data are sequences of measurements over time describing the behavior of systems. Time series analysis has become increasingly important in monitoring systems health and performance. As the system behavior changes over time due to external events and/or internal modifications, the problem of identifying the locations of these changes, referred to as Change Point Detection (CPD) has quickly drawn researchers attention. The CPD problem has been widely researched during the last three decades (4; 18; 13; 6; 16; 9) and it has been applied to several fields such as financial market analysis, medicine, climate science as well as system monitoring. The first methods found in the literature for CPD compared probability distributions between two consecutive intervals in a time-series, and alarmed if the difference became significant. Among them we find the cumulative sum algorithm (4) or the change finder for auto-regressive processes (18). Another line of research focuses on subspace identification (12), where the time series is modeled using a linear state-space and a change is identified using the model parameters.

Since all these methods make strong assumptions on the distributions, a need arises for more generic solutions and non parametric algorithms, such as direct density estimation methods. Unfortunately, these methods suffer from the curse of dimensionality (17) and are not applicable to real life problems. To overcome this challenge, one possible solution is to estimate the ratio of densities between two successive window without computing the densities themselves. This is achieved by going through the estimation of a probability divergence metric such as Kullback-Leibler in (16) or the Person divergence (13). Such methods proved to be quite successful. Another line of research focuses on Kernel two sample test (8), where the kernel trick is used to evaluate mean discrepancy of two samples in a Reproducing Kernel Hilbert Space. For example, Harchaoui et al. (10) introduced a test statistic using the maximum kernel fisher discriminant ratio. More recently, Chang et al. (6) proposed a way to learn an optimal kernel representation for CPD by using an auxiliary generative model.

Nonetheless, although more general, these models still assume that the process is time independent. However, very often, a time-series follows a seasonal behavior. Seasonality is defined as the tendency of a time-series to exhibit behavior that repeats itself every fixed period of time. The term season is used to represent the period of time before behavior begins to repeat itself. Detecting change in the seasonal pattern of a time-series is critical for many applications such as service mon-

itoring or climate change detection (5; 15). In some cases, it requires a totally different approach than regular CPD solutions. For example, Figure 1 shows the CPU utilization metric values recorded over a period of six days for some server machine. An event occurs every day at 10AM, representing some background process that is important for the system. A forecasting system should adapt its predictions to take into consideration this event. If for some reason, this process was moved to 4PM, the forecasting system is expected to detect this change and adjust the forecast values accordingly. Some information about the location of the peak in each period has to be taken into account in order to detect this kind of change.

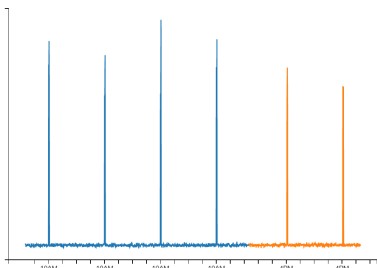

Figure 1: Example of change in process execution. On the first four days (blue section), the process is running at 10AM. During the two last days (orange section), it ran at 4PM.

The regular algorithms presented in this section would fail to detect such kind of change points. Indeed: (a) Existing parametric models for CPD compare the distribution between two windows by looking at some statistics. Here, we will not observe a persistent statistical change, as the spike still occurs. (b) Kernel based methods assume that the samples are generated $i.i.d$ (10; 13; 6). Hence, they will consider every seasonal spike as anomaly instead of including them to the model.

In this paper, we introduce a CPD variation for seasonal time series. The issue of seasonal CPD has also been addressed by Lund et al. (14) where they developed a test for periodic and auto correlated time series. However, this test is based on predefined statistics and focuses on median (level) change of the metric.

We present **ATR-CSPD**, a method which uses an Autencoder with Temporal Regularization for detecting Changes in Seasonal Pattern. Our contributions are the following:

- In section 2, we introduce a variation of the Change Point Detection (CPD) problem called Change in Seasonal Pattern Detection (CSPD) for seasonal time series, and explain why current solutions to CPD are inefficient at detecting them.

- In section 3 we describe **ATR-CSPD**, an unsupervised approach that is designed to capture essential information of each period in a seasonal time series. This is achieved by applying a time dependant regularization to the model's loss. We explain then how to use this model for CPD.

- Sections 4 and 5 present an extensive evaluation on both generated and real-life data. This benchmark exhibits how **ATR-CSPD** manages to detect new types of change points, undetected by regular algorithms.

## 2 PROBLEM FORMULATION

In this section, we start by presenting the simplified formulation of the Change Point Detection problem. Then we will introduce a more elaborated formulation for the detection of change points seasonal metrics, and illustrate the difference with an example. The simplified formulation assumes a single generative function behind the time-series values, and independence between the observations. Many common methods for detecting change points are applied based on this simplified version. For example, kernel density methods are based on the analysis of the single underlying function that generates the observations before and after the change (13; 6).

**Definition 1.** Change Point Detection (CPD):
Given a sequence of 1-dimensional observations $\{x_1, ..., x_t, ...x_N\}$ for which
$\forall i, \ x_i \in \mathbb{R}$. A change point $t_0$ is a point such that $\{x_1, ..., x_{t_0-1}\}$ are sampled $i.i.d$ from a distribution $F_1$, and $\{x_{t_0}, ..., x_N\}$ are sampled $i.i.d$ from the distribution $F_2$ where $F_1 \neq F_2$.

However, for a seasonal time-series the assumption that up to the change-point all samples are $i.i.d$ does not hold, since seasonality creates dependencies between the samples and their index in the period (i.e their phase). In the extreme, it might be that each observation within a period is drawn from an entirely different distribution. However, as we are targeting metrics that are generated by one origin system, a more plausible assumption would be that each data point $x_i$ is generated by a combination of a generative function of the time series and another generative function of its phase. For a seasonal time series with period size $p$, we define each point $x_i$ as $x_{jk}$ where $j$ is its period number, $k$ is its phase number, and $i = j \cdot p + k$.
We denote a single seasonal window of observations of size $p$ by $w_i$ where $w_i = x_{i1}, ...x_{ip}$. We can represent the original time series by grouping all its seasonal windows: $\{w_1, ..., w_t, ...\}, w_i \in \mathbb{R}^p$.

Denote $F$ as the generative function of the time series, $\{S_k\}_1^p$ as the phase-wise generative functions, and $x_{jk} \sim G_j$, where $G_j = S_j \otimes F$ and $\otimes$ can be an additive or a multiplicative factor (7). In additive seasonal models, the metric is explained by a weighted sum of the seasonal components $\{S_k\}_1^p$ and the generating function $F$. In multiplicative model, the sum is replaced by a multiplication. In our analysis and results we don't differentiate the two.

**Definition 2.** Change in Seasonal Pattern Detection (CSPD):
A change point in a seasonal pattern is a period number $j_0$ such that $\forall(k, j < j_0), \ x_{jk} \sim G_k$, and $\exists k, \ x_{j_0k} \sim G_k'$, where $G_k \neq G_k'$.

The difference between the two formulations can be demonstrated by the example displayed in Figure 1. It is clear that the time-series displayed in the chart represents a system that has altered its behavior. However, when considering the classical CPD problem formulation, the chart does not fall under the definition of a time series containing a change point as the overall distribution did not change. Considering the CSPD formulation, the change of the seasonal distribution component of both $S_{10AM}$ and $S_{4PM}$ distribution functions are identified as a change point in the time series. This observation suggests the CSPD is a generalization of the CPD problem. While CPD is centered around the cumulative values distribution parameters such as median and variance of the time series values, CSPD also focuses on the shape and proportions between the values observed in each period cycle.

Remark: Both problems can be extended to multi-dimensional time series and we are only presenting it in 1-dimension for simplicity.

## 3 AUTOENCODER WITH TEMPORAL REGULARIZATION

We introduce an algorithm that is able to detect changes in the seasonal pattern of a time-series. We start by using an autoencoder to capture the main pattern of each period in the time-series. The aim is to have close encoding (in terms of euclidean distance) for two periods that behaves similar, and different ones if there is an abrupt change between them. By having such a representation we can detect if there has been a change point by examining the euclidean distance between two adjacent encoded periods in the time series.

### 3.1 THE AUTOENCODER MODEL

Autoencoders are neural networks that attempt to copy its input to its output. It consists of an encoder function, that maps the input to an encoded version, and a decoder that performs the reconstruction. Here we are training an autoencoder to reconstruct fix-size windows of a time-series. Let $x_i \in \mathbb{R}^{d \times p}$, be the $i^{th}$ window of size $p$ in a $d$-dimensional time series, $f_{\theta_1} : \mathbb{R}^{d \times p} \to \mathbb{R}^{d \times q}$ our encoder function, $g_{\theta_2} : \mathbb{R}^{d \times q} \to \mathbb{R}^{d \times p}$ the decoder function and $n$ the total number of windows. In general autoencoder model, want to minimize:

$$\min_{\theta_1, \theta_2} \sum_{i=1}^n ||x_i - g_{\theta_2}(f_{\theta_1}(x_i))||_2 + \lambda(||\theta_1||_2 + ||\theta_2||_2) \tag{1}$$

$\theta_1$ and $\theta_2$ are a set of parameters that can be learn by gradient descent using back-propagation and $\lambda \in \mathbb{R}$ allows us to control the $L2$-norm of $\theta_1$ and $\theta_2$.

In **ATR-CSPD** a regularization term is added to this loss function, which will contribute to the detection of change points.

## 3.2 TEMPORAL REGULARIZATION

In order to encourage the network to generate similar low-dimensional representations, we introduce a new term in equation (1). The idea is to penalize the network for a difference between the encoding of two consecutive periods. The resulting loss function is given in (2).

$$\min_{\theta_1,\theta_2} \sum_{i=1}^{n} ||x_i - g_{\theta_2}(f_{\theta_1}(x_i))||_2 + \lambda(||\theta_1||_2 + ||\theta_2||_2) + \gamma \sum_{i=1}^{n-1} ||h_{i+1} - h_i||_1 \tag{2}$$

We refer to the last term as temporal regularization as it applies to neighbours period in the time-series. Here $\gamma \in \mathbb{R}$ allows us to control the strength of this regularization and $h_i = f_{\theta_1}(x_i)$.

To understand its effect we take the following examples. Figure 2a shows a one-dimensional, weekly seasonal, time-series. On it we run two autoencoder models, the first one optimizing equation (1) and the second one equation (2). We set $p = 288$ meaning that we consider the period as daily (the metric is recorded every 5 minutes), and expect to observe changes in the weekends. The additional parameters are set to $\lambda = 0.00005$, $\gamma = 0.001$ and $q = 18$. We observe that, although the regular autoencoder doesn't get rid of the major anomalies in the data (Figure 2b), the temporal regularized model creates a generic pattern for the weekdays and for the weekends (Figure 2c). According to this result, we can infer that the low dimensional representations are similar within the weekdays and within the days of the weekends, and we can use clustering to identify breakpoints. In Appendix C, we show the results of a Principal Components Analysis (PCA) applied to the encodings generated by both algorithms, which confirms this hypothesis.

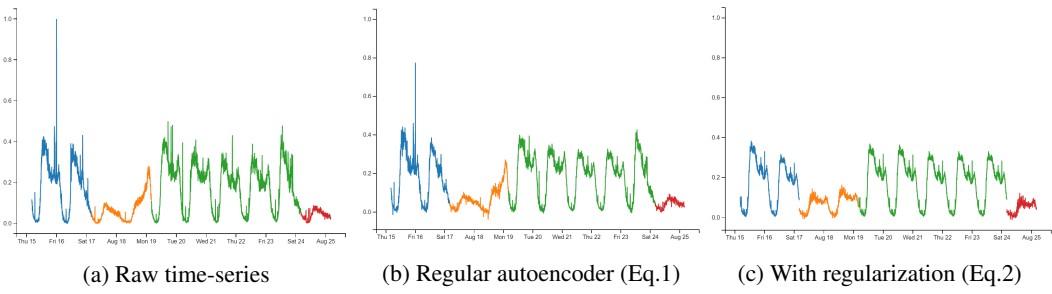

(a) Raw time-series      (b) Regular autoencoder (Eq.1)      (c) With regularization (Eq.2)

Figure 2: Given the time-series (2a), we run a regular autoencoder model (2b) and a temporal regularized version (2c). Even if it drops some anomalies, the regular autoencoder does not get rid of the minor differences between different days. However, using temporal regularization, the patterns for the week days (blue and green sections) are very similar, as well as the one for the weekends (orange and red).

## 3.3 LOCATING THE CHANGE POINTS

Once every window is mapped to an embedding in a reduced space, finding change points becomes much easier. We aim to find groups of similar windows, based on their location in the space. A change points will be detected if two consecutive points do not end up in the same partition.

To do that we choose to use the well known k-means clustering algorithm (11). In order to find the best number of clusters, and thus the number of change points, we use the silhouette score. It compares the mean pairwise distance of points in the same cluster ($a$) with the mean distance of each points to the nearest cluster ($b$): $s = \frac{b-a}{\max(a,b)}$

We run k-means iteratively on candidates number of change points, and select the one with the best silhouette score. If the received score is larger than a specified threshold (which is a hyperparameter of the model), we select the resulting partition, otherwise we consider that there were no changes.

## 4 EXPERIMENTS

In order to illustrate the difference between regular CPD algorithms and **ATR-CSPD**, we start by generating data containing different types of seasonal or non seasonal change points. Then we run some experiments on several real-life seasonal data sets. We present the results in the Smart meters in London (3) data set, the NYC taxi data set (2) and on time-series taken from Azure monitoring. In the experiments, we assume that for seasonal time-series the length of the period is known (the inference of the season is outside the scope of this article) .

### 4.1 NETWORK ARCHITECTURE

In all the experiments we use a similar network architecture. The encoder function $f_{\theta_1}$ is a 3-layer feed-forward neural network. Each layer consists of a linear function and a hyperbolic tangent function. Equation (3) shows how the layer output $z$ is computed from its input $x$. The decoder is a 2-layer neural network, with a similar activation function.

$$y = Wx + b, \quad z = \frac{e^y - e^{-y}}{e^y + e^{-y}} \tag{3}$$

The shape of $W$ in the layers determines whether it increases, decreases or leaves unchanged the dimension of the output. In **ATR-CSPD**, we reduce the dimension during the encoding phase and increase it back when decoding. This way, only the main information for the reconstruction will be stored in the encoding.

The initial window is always of the size of one period. For example if we have a daily seasonality and the data is recorded every 30 minutes, we will have a window size of 48. Then it is divided successively by 4, 2 and again 2 in the encoding phase, and multiplied by 2 and 8 in the decoding layers. The architecture is displayed in Appendix B.

Depending on the experiment, we use a learning rate that ranges from 0.005 to 0.05. The value of the parameter $\lambda$ in the loss function (2) is set to 0.00005 or 0.000001 and $\gamma$ varies from 0.001 and 0.00001.

Note that the time-series are first scaled to range between 0 and 1 using min-max scaling.

### 4.2 EVALUATION ON GENERATED SET

#### 4.2.1 SIMULATED SET WITH CHANGE POINT

For each category of change point, we generate 20 random time-series and keep track of the location of the change point inserted to the data. For all the seasonal time-series we fix the period length to 288. The first type is regular change points from non seasonal time-series. They can be a change in the mean, the variance or both on a white Gaussian noise without (category A), and with random anomalies (category B).
Then we generate time-series with seasonal spikes and white Gaussian noise and search for two different types of change points: a change in the height of the spike (category C), and a change in the position of the spike. (category D)
Finally, we consider seasonal time-series that alternate between a quiet state and an active state, with higher values and higher noise. On them we insert the following kind of change : A change in the active period height (category E), a change in the quiet period height (category F) and a change in the active period length (category G). We also generate 120 samples from those categories without changes in order to evaluate the false detection.

Samples for each category are drawn in Appendix A, along with more details about the implementation of the generating process.

#### 4.2.2 RESULTS

We compare the results for **ATR-CSPD** with two non-parametric change point algorithm: **KCpE** introduced by Harchaoui and Capp (9), and **RDR** presented by Liu et al. (13). We choose to use

a regular gaussian kernel: $K(x, x^{'}) = exp(-\gamma ||x - x^{'}||^2)$. The bandwidth parameters which are required for **KCpE** were set according to the "median trick" (8). For the **RDR** model, we follow Li et al. by adjusting the bandwidth and the regularization parameter at each time step. Finally, we run KCpE on deseasonalized data. Thus means, that for the seasonal samples, we compute the mean for each period and remove it from their original values. This is a common procedure for obtaining stationarity in seasonal time-series. We refer to this model as **DS-KCpE**.

Each model $F_i$ runs on a time-series $x_j$ and returns a set of candidate change points $\hat{t}^{ij}$. A change point $t_0^j$ in $x_j$ is detected by $F_j$ if

$$\hat{t}^{ij} = 1 \text{ and } |t_0^j - \hat{t}_0^{ij}| < 288. \tag{4}$$

Note that even if every model is able to detect multiple change points on a time-series, our generated samples all have at most one.
The detection for each category is equal to the number of change points detected divided by the category size. The false positive rate for $F_i$ is the ratio in time-series without change on which $F_i$ found at least one CP.

The results are summarized in Table 1:

| Category | KCpE | ATR-CSPD | RDR | DS-KCpE |
|---|---|---|---|---|
| A | **90%** | 40% | 50% | **90%** |
| B | **80%** | 0% | 15% | **80%** |
| C | 0% | **100%** | 20% | 0% |
| D | 0% | **90%** | 5% | 25% |
| E | 55% | **75%** | 50% | 55% |
| F | **100%** | 95% | 80% | **100%** |
| G | 55% | **80%** | 60% | 55% |
| False positives | 0.8% | **0%** | 2.5% | 0.8% |

Table 1: Results on generated set

The results clearly show that regular CPD approaches fail to detect most changes in the seasonal pattern when it comes to a seasonal spike. They also demonstrate that **ATR-CSPD** performs at least as well as classical algorithms on detecting change points in seasonal time-series. Indeed, in categories C and D, the impact of the changes on the kernel mean estimate of the windows is minimal or nonexistent, which explains why **KCpE** and **RDR** hardly detect them. We can notice that using decomposition of the seasonal component helped **DS-KCpE** for category D, still the detection is far from **ATR-CSPD**. A similar explanation can be derived for the next categories. Except for the category F, the changes are made only on a minor section of the periods and the statistical impact is smaller.
It worth noticing that **ATR-CSPD** performs badly on categories A and B. For a very noisy gaussian data, since getting a minor decrease in the MSE requires a high increase of the temporal regularization term, the autoencoder will not create a change in the encodings and the detection will be bad. In other cases, where there are too much anomalies on one period, it decides to learn the anomalies to decrease the MSE which might create a wrong detection of change point.

### 4.3 LONDON ELECTRICITY DATA SET

The data records the energy consumption of 5567 households in London between November 2011 and February 2014. For each time-series the consumption in kWh/hh is saved every 30 minutes. We assume that, in general, we will observe a weekly seasonal pattern, with similar weekdays and different weekends. Of course this assumption is not always true, for example retired people will probably exhibit a more monotonic week but the results will support this assumption.

First we preprocess each entry in the dataset by aggregating every 30 minutes of the week within the timeseries. Meaning that for a specific timeseries, we retrieve all the Mondays 8:30 AM and take the mean. Doing it for every recorded timestamp, gives us a new timeseries representing the average week for a household.

Then we run **KCpE** and **ATR-CSPD** on this new set. As **RDR** is quite slow, and we could not run it on this set. We say a model correctly detected a change if it detected changes only after Friday 00:00. Any other change is considered as false positive.

In Table 2 we present the precision and recall of the two models using the previous definitions. At a pretty similar precision level, we can see that our algorithm outperforms the baseline model. Figure 3 displays a few examples of unique detection of **ATR-CSPD**.

| Model | Precision | Recall |
|---|---|---|
| **KCpE** | 68% | 47% |
| **ATR-CSPD** | 67% | 54% |

Table 2: **ATR-CSPD** and **KCpE** on the London electricity set

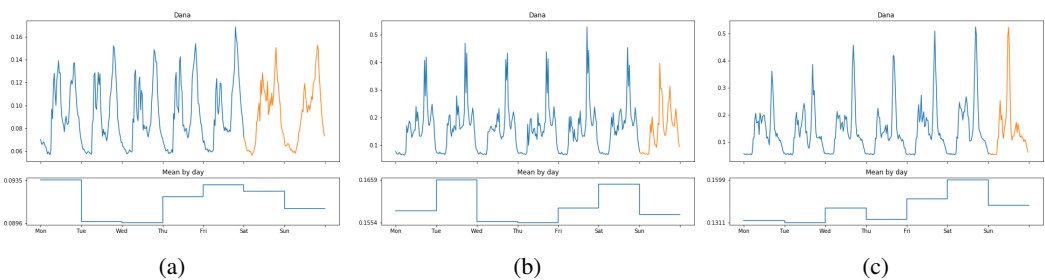

|     (a)     |     (b)     |     (c)     |

Figure 3: Electricity consumption of three households on which **ATR-CSPD** alone found a difference on the weekend. In all of them, it is not reflected by a drastic change in the mean. Top charts: electricity consumption. Bottom charts: the mean consumption of each day. In orange: periods recognized by **ATR-CSPD** as different

We can observe that the weekends, in those cases, is characterized by a different shape whether the level (mean) changes seem to occur at different places in the time-series. In Figure 3a we can easily distinguish the working hours of the weekdays, whether on the weekend no drop is observed. Looking at 3b, we notice that a peak of activity occurs every day. On Sundays, however, it happens at 12:00 PM instead of 4:00 PM days and **ATR-CSPD** spotted the change. Finally in 3c, we can observe that our model detected a change in the pattern on Sundays that did not drastically impact the mean.

## 4.4 NYC Taxi Dataset

The city of New York provides every month a dataset containing a lot of information about taxi trips (2). Taken from the kaggle Numenta Anomaly Benchmark repository (1), the current dataset consists of an aggregation of the total number of taxi passengers into 30 minutes buckets between July 2014 and February 2015.

The results of **ATR-CSPD**, compared with the two baseline models introduced in section 4, exhibit well the differences between regular Change Points and Changes in Seasonal Patterns. Looking at Figure 4 we, in fact, observe that although **KCpE** and **RDR** detects many level change, whether **ATR-CSPD** identifies only seasons in which the regular weekly pattern is broken. Looking at the dates of Figure 4b, one can notice that the weeks match respectively: the $4^{th}$ of July, Thanksgiving and Christmas holidays. Being able to identify non-regular seasons and to treat them separately could drastically benefit to a forecasting algorithm.

## 4.5 Azure Monitor data

Azure Dynamic Threshold (DT) Monitor is a service provided in the Azure cloud for monitoring a vast range of services and usage metrics. For this purpose, DT analyzes the past observable history of metric values and creates a baseline that describes the normal behavior expected for this metric. It later uses this baseline as a forecasting method. We used the training stage (analysis of past data) and run, as before, **KCpE**, **ATR-CSPD** and **RDR** to try to detect change point that happened in

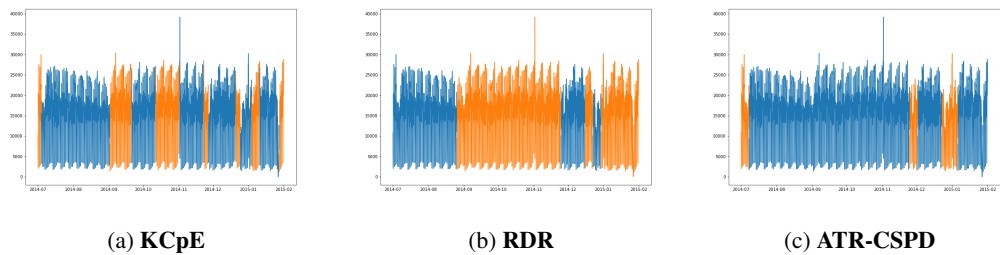

(a) **KCpE**                    (b) **RDR**                    (c) **ATR-CSPD**

Figure 4: **KCpE**, **RDR** and **ATR-CSPD** on NYC taxi dataset. Although **KCpE** (a) and **RDR** (b) are looking for a level change, **ATR-CSPD** (c) finds only changes in the pattern. They correspond to the weeks of the $4^{th}$ of July, Thanksgiving and Christmas holidays respectively.

those time series. We selected random 958 daily seasonal time series for this purpose. The values are recorded either every 5-15-30 or 60 minutes, and we use a total record of 10 days.

As there is not any label for this data, we will focus on the differences in the change points detected. We tune the three model to find change points in 25% of the time-series. Among the changes found by **ATR-CSPD**, 40% are not reported by the others. We present in Figure 5 a few examples of changes only detected by our model.

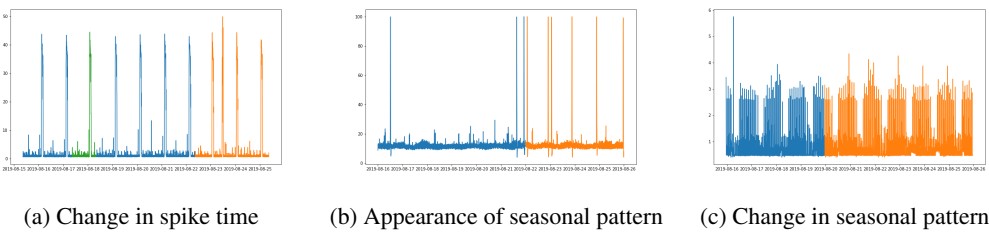

(a) Change in spike time          (b) Appearance of seasonal pattern          (c) Change in seasonal pattern

Figure 5: Three types of seasonal changes only detected by **ATR-CSPD**

In 5a blue spikes occurs between 2:40 and 2:45, in the green section, for some reasons it occurs at 1:35, an hour before usually. Finally in the orange section the spikes all occurs between 0:55 and 1:05. While monitoring the production system detecting such kind of update in a process could save false alarms while still allowing quick detection of anomalies.
In 5b, a seasonal pattern appears at the end of the time-series and **ATR-CSPD** managed to find the change points.
Finally in 5c, we can notice that even a slight change in a complex seasonal pattern can be detected as long it is repeated enough time.

## 5 CONCLUSION

We propose **ATR-CSPD**, a new deep learning-based algorithm that detects changes in seasonal pattern of a timeseries. The model combines an autoencoder network with a clustering algorithm, and is able to identify changes from different real-world applications. Evaluations on multiple benchmark datasets illustrate the difference between our new approach and existing CPD methods. They also demonstrate that **ATR-CSPD** outperforms other models in detecting specific types of change points and could be used to improve the efficiency of time-series forecasting in many applications.

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

# A   GENERATED DATA SAMPLES

## A.1   PLOTS

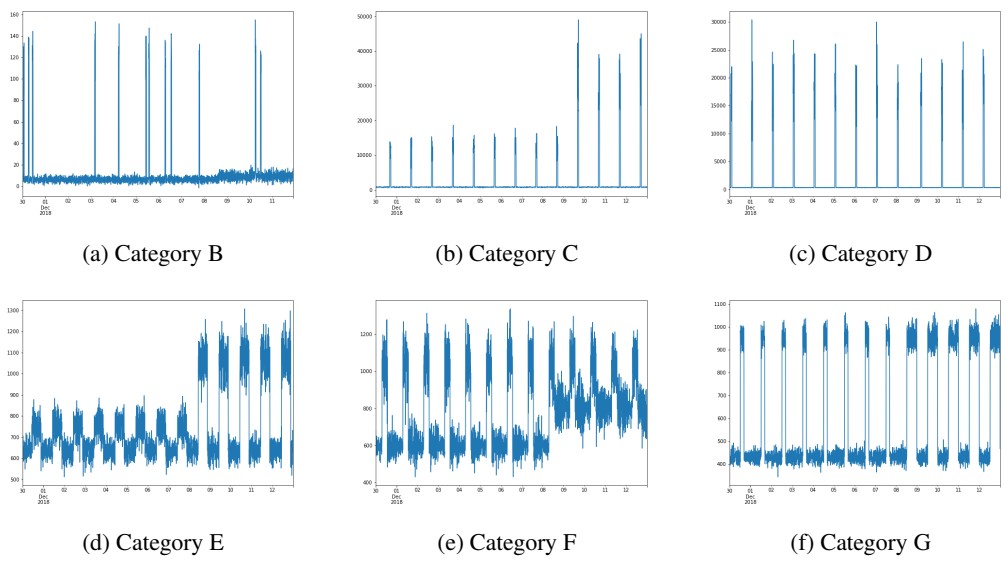

Figure 6: Samples from the generated dataset

## A.2   IMPLEMENTATION DETAILS

In this section, we explain, category by category, how samples in section 4.2 are generated.

**Category A**: Samples are following a gaussian distribution with a mean between 0 and 10 and a standard deviation between 1 and 10. Then we randomly choose whether a change will occur in the mean, in the standard deviation or in both. Finally, again at random, we define the parameters of after the change.

**Category B**: We use the same process as category A but we add some random spikes (anomalies). The spikes are generated with at a random frequency, there will be between 1 to 60 spikes in the time-series. Their height is also generated randomly, from 2 to 10 times the max value of the gaussian samples. Same for their length, between 5 and 15 points. At the end, we add some gaussian centered noise with a standard deviation that depends on its height.

**Category C**: We generate some regular gaussian samples with a random mean and variance. Then we add the spikes as previously but make them repeat every 288 points. For the last 4 spikes, we increase the height by a random factor (between 1.5 and 2).

**Category D**: The process is similar to the generation of samples of category C, but we create a random change in the periodic spike time and not in the height.

**Category E**: First we split randomly the period into three parts: a quiet part, an active part and a quiet part again. Then we generate some gaussian samples for the quiet part, with a random mean and a standard deviation that is determined by this mean. We do the same for the active part but make sure the mean is bigger. Finally, at a specific point in time, we randomly increase the mean of the active part.

**Category F**: We do the same as category E, but we randomly increase the mean of the quiet part and make sure it still stays lower than the active one.

**Category G**: Again, the same process as category E, but we do not modify the mean, we randomly change the split to make the active part longer.

## B   NETWORK ARCHITECTURE

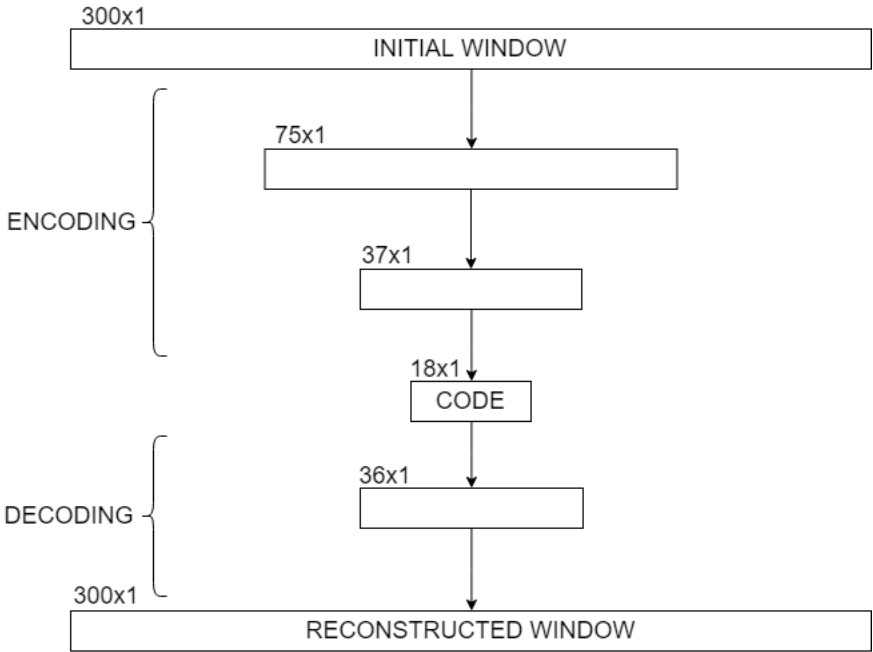

Figure 7: Network architecture for a window size of 300

## C   PCA ON THE ENCODINGS

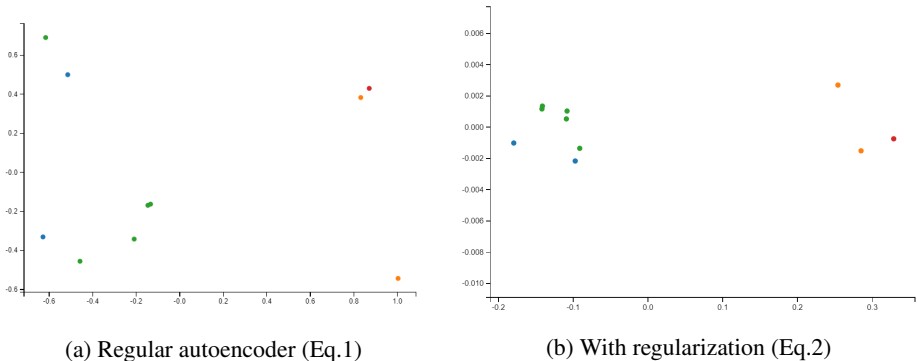

(a) Regular autoencoder (Eq.1)    (b) With regularization (Eq.2)

Figure 8: Results of a PCA applied to the encoding of Figure 2b and Figure 2c

