# OpenReview forum: "Detecting Change in Seasonal Pattern via Autoencoder and Temporal Regularization"
_ICLR.cc/2020/Conference — Reject_

### Official Review · AnonReviewer2 · 2019-10-16
**Official Blind Review #2**

**Rating:** 1

**Review:**

Summary: The paper raises an alarm that state-of-the art change-point detection methods in the ML literature do not handle important practical aspects arising in time-series modeling, namely seasonality. Indeed, methods designed to detect changing distribution under an i.i.d. setting can fail dramatically when the assumption is violated, when the change happens in the seasonal component.  The paper proposes to use an auto-encoder to find the "main pattern" within each seasonal window, and to use total variation penalty (l1-norm on the change) of the hidden state in the auto-encoder to encourage a smooth state-sequence which allow breaks.  They use k-means clustering to partition data-points, and detect a change-point if two consequent hidden states don't end up in the same cluster.

While the proposal is sensible and the paper is reasonably readable, I find the paper lacking in several respects, and recommend to reject it. My main concerns are
(a) novelty: despite the claims in the paper -- the importance of seasonality is well known and appreciated in time-series literature, and the proposal to look for changes in seasonality is fairly obvious when dealing with practical time-series. I would suggest to do a comprehensive literature search and re-evaluate the novelty of the paper.
I believe that recent ML papers e.g. kernel two-sample tests and such, focus on the i.i.d. setting and ignore seasonality (and other messy aspects of practical TS) -- as it is the more challenging statistical problem.
(b) The paper considers a setting where the time-series consists of a seasonal component and an i.i.d. component (combined additively or multiplicatively). It doesn't attempt to model any kind of stochastic dynamics -- e.g. at least a simple auto-regressive model instead of iid, and non-stationarity (trends) in the time-series. So despite aiming to look at practical time-series, the paper still considers a simplified model.
(c) The paper's presentation is often sloppy in language use, assumptions, mathematical details, and simulations and needs to be significantly improved to be considered for ICLR (or related ML conferences).

Detailed comments:
a) The references are severely lacking. There is an extensive literature in modeling time-series with seasonality and classical methods such as SARIMA (seasonal ARIMA), or exponential smoothing can track the evolution and changes in seasonal components. Various nonlinear DL-approaches to TS with seasonality have also started to appear. Once time-series is decomposed into trend, seasonal and stochastic part (using any linear or nonlinear or deep model), it is straightforward to apply anomaly detection algorithms to each component separately. Please take a look at e.g. https://anomaly.io/blog/index.html (from salesforce.com), to see practical change-point or anomaly detection in time-series in practice which does pay attention to seasonality. Also papers by Rob Hyndman pay close attention to seasonality, see e.g. https://otexts.com/fpp2/.
"Changepoint Detection in Periodic and Autocorrelated Time Series", https://journals.ametsoc.org/doi/full/10.1175/JCLI4291.1
https://cran.r-project.org/web/packages/trend/vignettes/trend.pdf  (which has a section on seasonal change-point detection)
Harvey, Koopman, Penzer, "Messy Time Series: A Unified approach", Adv. in Econometrics, Vol. 13, pp. 103-143., https://www.stat.berkeley.edu/~brill/Stat248/messyts.pdf
Perhaps there's relatively less focus on these practical details of change-point detection in recent ML literature and the focus is on the stochastic component, as it is the most challenging for prediction. The use of l1-norm of differences in time-series to detect changes is a natural idea, and has been suggested many papers e.g.in: http://eeweb.poly.edu/iselesni/lecture_notes/TV_filtering.pdf,
"Time Series Clustering using the Total Variation Distance",
Stephen Boyd's trend filtering, https://web.stanford.edu/~boyd/papers/pdf/l1_trend_filter.pdf .

While I am not aware of a specific prior work on auto-encoder with temporal smoothness for CPD, most of the main ideas are well known, and in my view the contribution is very limited in novelty.

b) You're ignoring any memory or dynamics in the stochastic component of the time-series -- e.g. allowing something like a simple AR model rather than iid would be a good step. Detecting changes in the dynamics or correlation structure (temporal or cross-sectional) would make the paper more interesting.  Something closer to switching linear dynamical systems, see for example https://arxiv.org/abs/1610.08466.

c) The presentation has many issues in language / math / simulations and needs to be improved:

1. The setting is not described clearly / formally -- are you trying to detect change-points online or offline, what assumptions are you making on the segments after removing seasonality -- are these just iid / stationary, can they include trends, outliers, e.t.c.
2. Baseline methods for detecting seasonal patterns are naive -- clearly applying methods that are not aware of seasonality will fail when there is strong seasonal components. There is one basic attempt at removing the seasonal component by averaging, and applying iid kernel CPD methods -- where it does help.  I believe doing something a bit more realistic (like doing a seasonal decomposition) will make the baselines much stronger.
3. Citation format is inconsistent with ICLR.
4. ATR-CSPD is undefined in the abstract.
5. Intro:  i, j, k notation inconsistent -- you seem to use i both for i = j*p + k, and also to refer to window id.
6. What is a "generative function" of time-series? Do you mean the pdf / cdf? What do you mean by a product of generative functions (which is additive or multiplicative), do you mean adding / taking products of random variables coming from independent distributions? What do you mean that you do not differentiate between additive / multiplicative? Do you claim to handle both within the same model?
7. Definition 2 -- do you look for x_jo,k ~ Gk',  or x_j for j> j0 ~ Gk'?
8. You claim a multi-variate extension is easy -- but is it? How would you tackle e.g. changes in correlation structure?
9. "Autoencoders attempt to copy input to output" - isn't this trivial by using an identity function? You should mention some compression / bottleneck as well.
10. How do you optimize the total-variation (l1-norm) penalty in your formulation? Just throw it into SGD in keras?
11.  The discussion in 3.2. is confusing -- you talk about weekly series, but use daily-seasonality, however you then describe detecting weekdays vs. weekends? How can you associate separate weekends without a weakly seasonal model?
12. London electricity data-set -- why do you average all weeks within the time-series to find average customer week? This was very surprising. Don't you loose most of the interesting anomaly data this way?
13. Figures are not explained well. While there's nice use of color -- it's often hard to understand what is the description pointing at.

typos: Person -> Pearson,  Autencoder -> Autoencoder, and many others.


**Experience Assessment:**

I have published one or two papers in this area.

**Review Assessment: Checking Correctness Of Derivations And Theory:**

N/A

**Review Assessment: Checking Correctness Of Experiments:**

I carefully checked the experiments.

**Review Assessment: Thoroughness In Paper Reading:**

I read the paper thoroughly.

---

### Official Review · AnonReviewer1 · 2019-10-24
**Official Blind Review #1**

**Rating:** 3

**Review:**

This paper proposed a new model for change point detection, using autoencoders with temporal regularization, in order to impose temporal smoothness in the latent codes. To motivate this new model, the authors also provided a toy example to show how the abnormality in a time series is removed in the reconstructed signal using this additional regularization term. Experimental results were provided to support the proposed new model.

I have a few concerns about some technical details of the paper, as explained below:
1) The paper motivated the new model with difficulty in detecting change points in seasonal time series. However, the proposed model with the temporal regularization is not directly related to the seasonality or periodicity of the input data. It is more related to the smoothness of the latent code. Hence it seems to me that there is a slight disconnection between the motivation and the actually proposed model. It would be nice if the authors can provide more intuitive explanation on why the temporal regularization can handle well change point detection in seasonal temporal series.

2) The temporal regularization proposed in this paper is very similar to the total variation penalty used extensively in statistics and image processing. It would be nice if the authors can make a connection between the two. For example:
Harchaoui, Z., & Lévy-Leduc, C. (2010). Multiple change-point estimation with a total variation penalty. Journal of the American Statistical Association, 105(492), 1480-1493.
Beck, A., & Teboulle, M. (2009). Fast gradient-based algorithms for constrained total variation image denoising and deblurring problems. IEEE transactions on image processing, 18(11), 2419-2434.

3) In the experimental section 4.3, the authors mentioned that "In Table 2, ... we can see that our algorithm outperforms the baseline model". However, in Table 2 the precision of the proposed model (67%) is lower than the baseline model (68%). Hence it is not obvious to the reader that the proposed model outperforms the baseline.

4) In the appendix B, the authors explained the architecture of the autoencoder used in the paper. I wonder why the authors chose an asymmetric structure between the encoder and decoder, as most autoencoders have a symmetric structure.

**Experience Assessment:**

I have read many papers in this area.

**Review Assessment: Checking Correctness Of Derivations And Theory:**

I assessed the sensibility of the derivations and theory.

**Review Assessment: Checking Correctness Of Experiments:**

I assessed the sensibility of the experiments.

**Review Assessment: Thoroughness In Paper Reading:**

I read the paper at least twice and used my best judgement in assessing the paper.

---

### Official Review · AnonReviewer4 · 2019-11-04
**Official Blind Review #4**

**Rating:** 3

**Review:**

The paper investigates the important problem of detecting changes in seasonal patterns, and proposes ATR-CSPD to learn a low-dimensional representation of the seasonal pattern and then detects changes with clustering-based approaches.  ATR-CSPD achieves improved results on part of synthetic and real-world datasets.

The paper may not have enough contribution to be accepted due to the following key concerns:
  - The proposed model is not quite novel, and the design needs more justification.
  - The empirical results are not strong enough to show the effectiveness of ATR-CSPD.

# Model design

The idea of using auto-encoder with temporal smoothing to learn a low-dimensional representation of time-series need more justification.
- What are the main intuitions of using an auto-encoder? e.g., removing anomaly or denoising. Why will it be easier for the model to detect the changes on the reconstructed time-series?
- The temporal smoothing makes adjacent periods similar to each other. However, this may have side effects like low recall.  For example, in Figure 2(a), the pattern in Aug 17th (Sat) and that in Aug 18th(Sun) can possibly be different (i.e., a change in seasonal pattern), while the difference is smoothed out by the temporal regularization.  Is the model sensitive to the regularization, e.g., $\lambda$? Why L1 regularization instead of L2 is used?  It will be helpful to provide more justification/intuition of the model design.
- Why only the smoothness regularization between adjacent seasons is used? Other potential regularization includes penalizing the difference between the same phase in different seasons.

# Assumption and limitation
The proposed method requires the seasonal period being provided, and also requires a large number of hyperparameters being specified, e.g., 1) the threshold of silhouette score, 2) the hidden representation dimension $q$, 3) the regularization coefficient, 4) $\gamma, \lambda$,  5) hyperparameters for constructing the encoder/decoder and 6) training the models.

Regarding the threshold of the silhouette score in the clustering step, is setting this hyperparameter easier than the number of clusters, i.e., the number of changing points?  Is ATR-CSPD sensitive to this parameter?  How this hyperparameter is tuned? Having too many hyperparameters (that are potentially non-trivial to set/tune) may make the proposed method less robust.

# Experimental results:
According to the results in Table 1, ATR-CSPD is mainly better at detecting Category C/D/E change points, which are mainly caused by changes of height/position of the spike.  However, it performs either similarly or worse than the other baselines on other tasks.  Besides, the lack of ground-truth data on NYC Taxi dataset and the Azure monitor dataset makes it hard to evaluate the effectiveness of the proposed algorithm. Moreover, only uni-variate time series tasks are investigated.  These issues may limit the application domain of the proposed algorithm.

# Minor notation and presentation issues
- In Definition 2, does CSPD assume F is the same in $G_k$ and $G'_k$? If not, CPD might be a subset of CSPD.

- In Equation 1 and 2, $n$ is used to represent the number of observations, while in Definition 1, capital $N$ is used to represent the same concept.

- In Figure 2, it might be easier to understand if all the weekdays are drawn using the same color (blue or green) and all the weekends are also drawn in the same color (yellow or red).

**Experience Assessment:**

I have published one or two papers in this area.

**Review Assessment: Checking Correctness Of Derivations And Theory:**

I assessed the sensibility of the derivations and theory.

**Review Assessment: Checking Correctness Of Experiments:**

I assessed the sensibility of the experiments.

**Review Assessment: Thoroughness In Paper Reading:**

I read the paper at least twice and used my best judgement in assessing the paper.

---

### Official Review · AnonReviewer3 · 2019-11-10
**Official Blind Review #3**

**Rating:** 1

**Review:**

I am quite disappointed with the presentation and technical quality of the paper.

There are numerous grammatical errors that make the reading unpleasant. The mathematical notations are also inconsistent throughout different places in the paper.

The extensive literature of modelling time series with seasonality trends, both in the statistics and the machine learning community, is severely under-represented in the motivations and related works. Models like SARIMA have no mention in the paper.

The temporal regularization imposed in section 3.2, coupled with an autoencoder, does not seem very different from the state-space models and their more complex and recent variants that use multi-layered networks (a google search will provide plenty references).

The experiments, with many of the useful baselines missing, are equally unimpressive.

**Experience Assessment:**

I have read many papers in this area.

**Review Assessment: Checking Correctness Of Derivations And Theory:**

I assessed the sensibility of the derivations and theory.

**Review Assessment: Checking Correctness Of Experiments:**

I assessed the sensibility of the experiments.

**Review Assessment: Thoroughness In Paper Reading:**

I read the paper at least twice and used my best judgement in assessing the paper.

---

### Decision · Program_Chairs · 2019-12-19

**Decision:**

Reject

**Comment:**

The paper proposes ATR-CSPD, which learns a low-dimensional representation of seasonal pattern, for detecting changes with clustering-based approaches.

While ATR-CSPD is simple and intuitive, it lacks novel contribution in methodology. It is unclear how it is different from existing approaches. The evaluation and the writing could be improved significantly.

In short, the paper is not ready for publication. We hope the reviews can help improve the paper for a strong submission in the future.